# Brain Wearables: Validation Toolkit for Ear-Level EEG Sensors

**DOI:** 10.3390/s24041226

**Published:** 2024-02-15

**Authors:** Guilherme Correia, Michael J. Crosse, Alejandro Lopez Valdes

**Affiliations:** 1Department of Physics, NOVA School of Science and Technology, 2829-516 Caparica, Portugal; lourenog@tcd.ie; 2Segotia, H91 HE9E Galway, Ireland; mick@segotia.xyz; 3Trinity Centre for Biomedical Engineering, Trinity College Dublin, D02 R590 Dublin, Ireland; 4Global Brain Health Institute, Trinity College Dublin, D02 X9W9 Dublin, Ireland; 5Trinity College Institute of Neuroscience, Trinity College Dublin, D02 X9W9 Dublin, Ireland; 6Department of Electronic and Electrical Engineering, Trinity College Dublin, D02 Dublin, Ireland

**Keywords:** in-ear EEG, hearables, brain–computer interfaces, non-invasive, electroencephalography, EEG phantom

## Abstract

EEG-enabled earbuds represent a promising frontier in brain activity monitoring beyond traditional laboratory testing. Their discrete form factor and proximity to the brain make them the ideal candidate for the first generation of discrete non-invasive brain–computer interfaces (BCIs). However, this new technology will require comprehensive characterization before we see widespread consumer and health-related usage. To address this need, we developed a validation toolkit that aims to facilitate and expand the assessment of ear-EEG devices. The first component of this toolkit is a desktop application (“EaR-P Lab”) that controls several EEG validation paradigms. This application uses the Lab Streaming Layer (LSL) protocol, making it compatible with most current EEG systems. The second element of the toolkit introduces an adaptation of the phantom evaluation concept to the domain of ear-EEGs. Specifically, it utilizes 3D scans of the test subjects’ ears to simulate typical EEG activity around and inside the ear, allowing for controlled assessment of different ear-EEG form factors and sensor configurations. Each of the EEG paradigms were validated using wet-electrode ear-EEG recordings and benchmarked against scalp-EEG measurements. The ear-EEG phantom was successful in acquiring performance metrics for hardware characterization, revealing differences in performance based on electrode location. This information was leveraged to optimize the electrode reference configuration, resulting in increased auditory steady-state response (ASSR) power. Through this work, an ear-EEG evaluation toolkit is made available with the intention to facilitate the systematic assessment of novel ear-EEG devices from hardware to neural signal acquisition.

## 1. Introduction

Conventional electroencephalography (EEG) is an invaluable tool for assessing neurological disorders, with measurements performed in controlled clinical environments, utilizing full cap systems over the scalp with wet electrodes that provide low impedance for quality measurements with high temporal resolution. However, researchers and clinicians have been interested in measuring EEG outside the laboratory for better assessing neurological disorders [1].

The development of technology has led to the evolution of ambulatory EEG, resulting in research on smaller EEG measurement options. This has given rise to a new category of wearable EEG devices that are wireless, have aesthetic designs limited to only the area around the head, and use dry electrodes. The convenience of these devices enables individuals to incorporate EEG measurement into their everyday lives, making it easier to assess certain medical conditions such as epilepsy and sleep disorders and allowing for brain-computer interface (BCI) applications outside of a laboratory environment. While these types of devices are commercially available and their capabilities acceptable for some current applications, the overall design of these wearables still constitutes a barrier to daily usage, being uncomfortable over long periods of time and making it obvious when a person is utilizing such an EEG device [2,3].

Quoting Looney et al., the next generation of wearable devices must be “discreet, unobtrusive, robust, user friendly and feasible”. An in-ear EEG approach checks all these boxes, trading the wide coverage across the scalp with an inconspicuous EEG recording ability based on the ear [4]. The ability to acquire reliable brain recordings from inside the human ear is critical to accelerating the development of next-generation EEG-enabled earbuds. While ear-EEG is one of the best candidates for consumer BCI, being referred to as “beyond wearable” [2], real-world brain recordings are affected by multiple environmental factors that are not present in the laboratory. A systematic, well-established characterization of in-ear EEG hardware and signal quality is crucial to understand the limitations and applications of newly developed devices.

When characterizing a novel EEG system, it is important to be able to evaluate all components of the system. This includes the biological neural signals, derived from validated EEG paradigms like the alpha block or the various event-related potentials (ERPs) and the hardware and signal chain components (electrical and mechanical), which require testing in a controlled setting through a test-bench or phantom model [5].

In many areas of research and instrumentation, phantoms are utilized for testing, validating, and calibrating acquisition systems, namely in medical imaging like positron emission tomography (PET) and magnetic resonance imaging (MRI) scans where standardized phantoms are available [6]. In EEG, a standardized phantom does not exist, although many prototypes from different materials and designs have been proposed.

A common approach used by research groups to test novel EEG systems or new types of electrodes consists of using salt-doped ballistic gelatin (BG) head models onto which scalp-EEG sensors can be mounted for testing [7,8]. Nonetheless, phantoms can also be made from different organic (e.g., agar) or synthetic (e.g., carbon-doped thermoplastics) materials [9,10]. The basic principle for an EEG phantom prototype is to obtain a conductive physical model with the shape of the human head. This is usually achieved through image processing of anatomical scans (i.e., MRI or CT) and 3D printing of the resulting phantom’s cast for molding [11].

An EEG phantom allows for the playback of a previously recorded known EEG signal. This recognizable signal is referred to as the “ground truth”. Then, measurements can be made on the phantom for comparison with this known signal, which constitutes a way of characterizing the acquisition system without the need to account for the inherent variability and lack of repeatability of neural signals present when recording from human subjects, as well as identifying external sources of noise [12,13]. For a known previously recorded signal to be played out of the phantom, electrode antennas are driven inside it. These can be simple screws, exposed wire tips, or a coaxial cable to create dipoles [10,14,15]. The driven antennas can either be attached to the interior side of the scalp layer in hollow-shaped phantoms or be put in place during a phase of assembly where the constitution of the filling material allows for this procedure [10,16]. Additionally, EEG phantoms allow for a controlled analysis of electrode contact impedance measurements and signal noise floor characteristics [16,17].

Despite the existence and use of EEG phantoms for evaluating scalp-based EEG systems, there is a lack of literature for ear-EEG validation via appropriate phantom models. To the best of our knowledge, appropriate EEG phantoms for this purpose do not exist and all previous EEG phantoms neglect the structures of the outer ear and ear canal systematically.

The feasibility of measuring brain signals through ear-EEG devices has been validated in recent publications [4,18,19]. For an in-depth review of the field and overview of the technological state-of-the-art around ear-EEG we please refer to [20,21]. A commonality across the ear-EEG literature is that the validation paradigms utilized tend to be known ERP paradigms, namely, the alpha modulation (or alpha blocking) paradigm, the auditory steady-state response (ASSR), the steady-state visual evoked response (SSVEP), auditory evoked potentials (AEPs), visual evoked potentials (VEPs), and oddBall-type paradigms to elicit responses linked to higher processes of the brain, like the P300 and mismatch negativity (MMN) responses. Similarly, the interest of utilizing electro-oculography (EOG) measured by ear-EEG as a possible input for BCI applications has also been explored [19].

Given the predominance of this select group of paradigms utilized to derive ear-EEG responses and the lack of a dedicated phantom model suitable for ear-based systems, here we propose a complete ear-EEG validation toolkit to contribute to and expand the characterization of this technology through:A software framework (“EaR-P Lab”) that allows the user to readily make a validation test battery for the characterization of ear-EEG devices at the neural signal acquisition level.The design and prototyping of an ear-EEG-suitable physical phantom for systematic characterization of in-ear sensors, allowing controlled comparison of fit form factors for ear-EEG acquisition.

This contribution to the field aims to allow for a more reliable assessment of out-of-the-box ear-EEG devices by providing proper benchmarking tools for comparing systems.

## 2. Materials and Methods: Ear-EEG Toolkit Design and Validation

### 2.1. EaR-P Lab—Design and Validation

A custom application, programmed with nine commonly utilized EEG recording paradigms, was created for the assessment of ear-EEG devices at the neural signal level. This application was written in the Python programming language and consists of a collection of scripts (requiring a local Python installation), pre-synthesized stimuli, and other auxiliary files and folders wrapped in an usable graphical user interface (GUI), based on the Tkinter Python package. The stimuli delivery and precise experiment timings are handled via PsychoPy, a Python-based open source package for experiment control [22].

We dubbed this software script “EaR-P Lab” from the amalgamation of the words “ear” and “ERP”, as per the motivation of streamlining the measuring of ERPs (as well as other paradigms) from the ear. The organizational structure and main functionalities of EaR-P Lab are depicted in Figure 1.

When initialized, EaR-P Lab provides a user-friendly GUI to present the necessary stimuli and insert the relevant markers that delimit different paradigms or indicate the start of a given stimulus in the validation test battery. The marker’s timestamps are synchronized with the EEG data through the Lab Streaming Layer (LSL) protocol that handles the synchronization between the EEG data and maker streams through a local network (an EEG amplifier compatible with LSL must be used) [23,24]. The marker and EEG data streams can then be recorded with another software (i.e., LabRecorder) into a single *.xdf* file on the hard disk of the computer for offline analysis, as illustrated in Figure 2.

#### 2.1.1. GUI—Main Menu

When EaR-P Lab is launched, the main menu in Figure 3 appears. This menu gives the user access to the nine pre-loaded EEG acquisition paradigms, with the script automatically opening the LSL marker stream using the *pylsl StreamOutlet* function at startup. Eleven buttons are presented: nine that direct to the different paradigms and two (Change Settings and Change Markers) that allow users to modify specific experiment parameters. Some disclaimer information on how EaR-P Lab operates is also shown in this window. Refer to Appendix A for a full overview of the features and functions of each button and the settings menus.

#### 2.1.2. Stimuli and Trigger Latency

To accurately evaluate event-related potentials (ERPs), good synchronization between data and markers sent at stimulus onset (triggering) is crucial. The PsychoPy library allows sounds to be pre-scheduled with the high precision “PTB” settings and the *callOnFlip* method to start visual stimuli promptly, both used in EaR-P Lab [25]. However, unavoidable latency and jitter issues related to screen refresh rates and monitor syncing may still arise [26].

To measure the timing characteristics of EaR-P Lab, we used an amplifier-specific accessory (i.e., mBrainTrain Delay/Jitter (DJ) box). This accessory includes an audio input, output, and photo-diode sensor that allowed us to test the marker synchronization with the audio and visual stimuli. The device was placed at the latest stage possible in the audio delivery setup, with the photo-diode centered and 30 cm away pointing to the center of the screen.

During the early testing phase, we observed an increasing cascading effect on the latency delay when recording different blocks of transient responses within the same *.xdf* file, illustrated in Figure 4.

Our troubleshooting showed that to correct this effect, the data streaming from the amplifier software (mBrainTrain Streamer) had to be restarted after each EEG recording block. Hence, each block of a given paradigm must be recorded in a separate *.xdf* file to ensure reliable trigger latencies, as shown in Figure 5.

However, to the best of our knowledge, this may only apply to the mBrainTrain Smarting Mobi recording platform. We recommend testing the system’s latency before recording and having separate *.xdf* files for the AEP, VEP, EOG, and oddBall-type blocks to avoid latency synchronization issues.

Once our latency issues were resolved, the latency delay was measured for 1000 trials of both auditory and visual events, showing a mean delay of 0 ms (2.2 ms std./jitter) and 21 ms (2.8 ms std./jitter), respectively. The processing of the EEG paradigms accounted for these measured latencies. These values are specific to the laptop used to present the stimuli and are likely to vary between systems.

#### 2.1.3. Data Acquisition and Test Battery

In order to validate EaR-P Lab as a tool for assessing ear-EEG (and EEG in general), standard scalp-EEG data were acquired for each of the EEG paradigms. EEG data were collected from five test subjects with normal or corrected-to-normal vision and normal hearing (age avg. 27.0 ± 4.3 std.). A custom EEG cap with 12 channels was used for the benchmark condition. The cap uses conventional Ag/AgCl electrodes at the F3, Fz, F4, T7, C3, Cz, C4, T8, P3, Pz, P4, and Oz positions. The reference electrode (CMS) and ground (DRL/GND) were located at FCz and AFz, respectively. Conductive gel (Signagel, Parker) was applied to each scalp electrode, typically achieving impedances of <5 kΩ, measured by the mBrainTrain Streamer. EEG data were recorded using the mBrainTrain Smarting Mobi amplifier—capable of high-quality wireless recordings—at a sampling rate of 500 Hz with wireless data transmission, providing 24-bit resolution in the analog-to-digital converter (ADC).

A laptop (ASUS VivoBook 15, Windows 11 OS, Intel Core i7-10750H Processor, NVIDIA GeForce GTX 1650 Ti Graphics Card, 12 GB RAM) ran the essential recording software (mBrainTrain Streamer and LabRecorder) and stimulus presentation software (EaR-P Lab). Data were transmitted via Bluetooth from the amplifier secured at the back of the subject’s head to a USB dongle connected to the laptop. Auditory stimuli were delivered using research-grade ER-2 earphones fitted with foam ear tips (Etymotic Research, Inc., Elk Grove Village, IL, USA), shielded with copper tape to reduce interference. To ensure that the stimuli were audible without being distorted, an external sound card (TASCAM US-100) and digital-to-analog converter (DAC, FiiO Alpen2) were used (see Figure 6). The laptop and DAC were set to the maximum volume, with adjustments made to the sound card only. Subjects were instructed to adjust the volume to a comfortable, audible level. The laptop screen brightness setting was set to the maximum for visual stimuli.

EEG recordings were conducted in a dedicated sound-attenuated room with only the participant and the technician present. Participants sat in a comfortable chair and were instructed to relax and minimize motor and eye movements during recordings. Participants were given a brief explanation of the recording process at the start of the session, and detailed instructions were provided at the beginning of each task.

The EaR-P Lab EEG paradigms employed were as follows:**Resting State:** 4 min of resting-state EEG recording (Figure A1)**Alpha Block:** 4 min of eyes open/closed, 1 min per block (Figure A3)**ASSR:** 4 min of continuous auditory stimulation, 1 kHz carrier signal with 40 Hz amplitude modulation (Figure A1)**SSVEP:** 4 min of continuous visual stimulation, 10 Hz flickering radial checkerboard, subjects seated at a distance of 60 cm from the center of the screen with room lights turned off (Figure A2)**AEP:** 200 trials of discrete auditory events, 1 kHz pure tone of 200 ms duration with 10 ms rise/fall time, interstimulus interval (ISI) between 1200 and 1800 ms, total duration of 7–8 min (Figure A4)**VEP:** 200 trials of discrete visual events, pattern-reversal radial checkerboard of 500 ms with 500 ms ISI, total duration of 5 min (Figure A5)**AEP OddBall:** 200 trials of discrete standard/deviant auditory events (standard: 440 Hz pure tone; deviant: 880 Hz pure tone), 100 ms duration with 10 ms rise/fall time and 1200–1800 ms ISI, total duration of about 15 min**VEP OddBall:** 200 trials of discrete standard/target visual events (standard: blue square; target: red circle), 500 ms duration with 600–700 ms ISI, total duration of about 18 min, subjects instructed to respond to target with button press (Figure A6)**EOG:** 80 trials of discrete visual events, 500 ms duration dot movements with 1000–1600 ms ISI, total duration of about 10 min, subjects seated at a distance of 30 cm from the center of the screen (i.e., visual angle of 16.2°); subject’s head was stabilized using an adjustable chin rest and the monitor was centered with the subject’s eyes (Figure A7).

Each acquisition session took about 2 h to complete from start to finish, including EEG cap setup. Subjects were entitled to short breaks (2–5 min) at the end of each paradigm but were also allowed longer breaks when requested. For some subjects, the protocol’s order was not strictly followed.

#### 2.1.4. Data Processing and Statistical Analysis

All data preprocessing and analysis was performed via custom scripts in MATLAB (MATLAB (R2020a), The MathWorks Inc., Natick, MA, USA). EEG data were first highpass-filtered (Butterworth IIR), notch-filtered (50 Hz second-order IIR), and then lowpass-filtered (Butterworth IIR). Frequency-based paradigms (ASSR, SSVEP, Alpha Block) were bandpass-filtered between 1 and 100 Hz, and transient responses (AEP, VEP, MMN, P300, EOG) were bandpass-filtered between 1 and 20 Hz. To isolate the EOG blinks in the respective three-second window, these data were bandpass-filtered between 0.2 and 3 Hz. All EEG data were re-referenced to Cz.

ASSR and SSVEP responses were extracted by computing the power spectral density via Welch’s method using an 8 s window with 50% overlap, and the alpha block spectrograms were obtained by computing the short-time Fourier transform (STFT) using a 2 s window with 50% overlap.

All transient responses were baseline-corrected based on the 100 ms pre-stimulus interval and then averaged across trials. To obtain the MMN and P300 responses, the average standard cue waveform was subtracted from the average target cue waveform for each subject, with the standard events immediately before the oddBalls being considered [27].

To measure the signal-to-noise ration (SNR) of the ASSR responses, the power at 40 Hz was compared with the surrounding frequency bins, according to Equation (Equation 1) [18]:(1)$$40\;Hz\;SNR=\frac{P\;(40\;Hz)}{P_{\mathrm{average}}{\left(35–45\;Hz\right)}^{*}}\;,{\;}^{*}excluding\;40\;Hz$$

The same formula was adapted for the 10 Hz SSVEP responses as per Equation (Equation 2):(2)$$10\;Hz\;SNR=\frac{P\;(10\;Hz)}{P_{\mathrm{average}}{\left(5–15\;Hz\right)}^{*}}\;,{\;}^{*}excluding\;10\;Hz$$

An *F*-test was also conducted on these ratios using a one-way analysis of variance (ANOVA).

Equation (Equation 3) was used to calculate alpha power by quantifying the power ratio of the eyes-closed and eyes-open conditions between 8 and 12 Hz.
(3)$$R_{AM}\;=\frac{P_{\mathrm{average}}\;\left(Alpha\;Band_{\mathrm{Eyes}\;\mathrm{Closed}}\right)}{P_{\mathrm{average}}\;\left(Alpha\;Band_{\mathrm{Eyes}\;\mathrm{Open}}\right)}\;,\;Alpha\;Band\;\left(8–12\;Hz\right)$$

Significant modulations in the average ERP amplitude were detected using one-sample (two-tailed) *t*-tests and are indicated in each of the plots by the green highlighted segments (p<0.05, not corrected for multiple comparisons) [28].

EOG profiles were quantified by the amplitude and polarity of each movement type. For the EOG blinks, an amplitude factor between the averaged peak-to-peak amplitude of the intentional blinks over the regular blinks was established [18].

#### 2.1.5. EaR-P Lab Validation

The following section describes the scalp-EEG data used to validate EaR-P Lab’s EEG paradigms. Each validation analysis was conducted at both the typical scalp site for a given test (e.g., electrode 0z for a visual stimulus), as well as a location close to the ear (electrode T8) for ear-EEG validation.


*Alpha Block*


As seen in Figure 7, a distinct effect on alpha wave blocking (with suppression of the 8–12 Hz band occurring during eyes open) was achieved with the highest modulation at Oz (6.1 dB), dropping to 3.7 dB near the ear at T8.


*ASSR*


For the ASSR (Figure 8), the SNR at the 40 Hz modulation frequency was measured, with the highest mean SNR at P4 (9.9 dB), dropping to 4.7 dB at T8.


*SSVEP*


The SSVEP at the 10 Hz modulation frequency was most pronounced at Oz with a mean SNR of 11.0 dB (Figure 9), as well as harmonics of the 10 Hz fundamental frequency up to the eighth harmonic present. The SSVEP was weaker at T8, with a mean SNR of 7.5 dB, with harmonic responses only distinguishable up to 40 Hz.


*AEP*


The AEP response to the 1 kHz tone is shown in Figure 10, comprising three significant deflections corresponding to the P1, N1, and P2 AEP components. The largest deflection was around 100 ms with an amplitude of 5 µV, corresponding to the N1 component (positive due to re-referencing), and a peak-to-peak amplitude of 8.9 µV between the N1 and P2 components.


*VEP*


The VEP response to the pattern-reversal radial checkerboard is shown in Figure 11. At Oz, a significant component is present between 150 ms and 300 ms with a peak amplitude of 15 µV at 200 ms, corresponding to the P2 VEP component. At T8 (near the ear), the P2 component was not significantly different from zero, instead showing two smaller positive deflections of 2 and 3 µV at around 90 and 300 ms, respectively.


*AEP OddBall (MMN)*


The MMN response was the strongest at electrode Pz, shown in Figure 12. At 200 ms, the expected negative differential between the standard and deviant ERPs was observed with an amplitude of 2 µV and was also present at electrode T8 with a similar amplitude.


*VEP OddBall (P300)*


The P300 response to a target visual stimulus was better observed at P4 at around 280 ms with an amplitude of 6 µV (Figure 13). In contrast, the P300 response at T8 only peaked at 4 µV with that amplitude achieved at around 250 ms from the onset.


*EOG (Blinks and Saccades)*


While not cortical in origin, EOG responses to blinking and saccade movements were also assessed as possible inputs for BCI applications. EOG responses are naturally more prominent at frontal scalp electrodes such as Fp1 and Fp2, as they are near the eye muscles responsible for these movements. In our scalp-EEG validation analysis, we considered F3 and F4 as approximate locations for Fp1 and Fp2.

Figure 14 shows the peak-to-peak EOG amplitude at electrode F3 for hard blinking (8 mV) and soft blinking (1.5 mV) for an example subject, with a lower absolute peak-to-peak EOG amplitude at T8 (but a similar soft/hard ratio).

Figure 15 shows the saccade profiles in different directions. The saccades were assessed at around 200 ms after stimulus onset, where the separation between directions is greatest. Between T7 and T8, it is possible to see a 10 µV difference between the right (blue trace) and left (green trace) saccades, with their polarity inverting, depending on the side of the sensing electrode. The vertically oriented saccades are less differentiated, with only around 2 µV amplitude separating their traces.

### 2.2. Ear-EEG Phantom—Design and Validation

A physical test bench to evaluate ear-EEG devices was developed. The basis for the prototype was similar to the most common type of EEG phantoms: a mold that creates the proper shape, is fit for use with an ear-EEG device, and can be filled with a conductive mixture.

In contrast to most common EEG phantoms, this work focused on developing a compact design solely focused on the ear anatomy instead of a complete head-shaped phantom. We also set out to create a modular ear section for the mold, allowing for testing different ear form factors without having to rebuild an entirely new mold, as well as having a way to secure the signal antennas in place during the setting process of the filling material.

The phantom was designed using the Fusion 360 computer-aided design (CAD) software and then exported as an *.stl* file for 3D printing. The base design is an 18 cm modular cylinder split into two parts along its length with open ends. The bottom half has two longitudinal railings where the top fits, leaving two round holes centered in the middle for inserting antennas. The cylinder sides can then be closed with the lids. The top has two square openings for pouring of the conductive mixtures. A 0.1 mm gap was used for all fitting parts to account for the 3D printing resolution. The rendered prototype is shown in Figure 16, and the phantom prototype’s full dimensions are presented in Appendix B.

The inside of these lids can then accommodate the outer ear canal and concha shape to create an imprint on the material inside, where an ear-EEG device can be placed for testing. The current methodology used ear canal scans by a professional audiologist, digitally scanned on-site, and delivered as an *.stl* file (as shown in Figure 17), which can be imported as a mesh to Fusion 360 and easily fused with the lid mesh by using the *Mesh Menu - Modify - Combine - Join Operation*.

Ear canal scans were centered on the lid’s mesh with the concha cymba outline parallel to the mesh’s top/bottom. Meshes intersected to an adequate depth as long as the ear canal and concha structures were visible and protruding, similar to Figure 18.

The ear-EEG phantom prototype was 3D-printed on a Prusa i3 MK3 3D printer using polylactic acid (PLA). PLA was chosen due to its ease of use, fast printing times, affordability, non-warping nature, and lack of post-processing requirements. The printing settings on the Prusa software were set to a 0.15 mm *QUALITY* printing resolution, using *Prusament PLA* filament with a default structural infill of 15%. The software automatically added necessary support structures where required. The combined printing time of the phantom body and lids was approximately 33 h. To minimize difficult-to-remove supports, the cylinder halves were printed vertically. If the phantom’s form-fitting factor needs to be changed, printing just a set of lids with the ear impression takes only 6 h. Figure 19 shows the different modular parts after printing and support removal.

#### 2.2.1. Phantom Assembly and Bulk Materials

To assemble the phantom, the antennas (pair of 3.5 mm AUX cables) were adjusted to fit the diameter of the cylinder holes by inserting one side of each cable inside PVC tubing and sealing it with duct tape. The cables were placed on the bottom half, then joined by the top and the lids with ear imprints on the side, holding the prototype together. Three strips of plumbing tape were used on the railing fittings, cables, and inner circumference of the phantom lids to prevent leaks, as in Figure 20. The phantom could then be filled with a conductive substance.

Here, three materials were tested with our design: agar doped with salt, BG doped with salt, and silicone doped with carbon fiber (CF), as a non-perishable option.


*Agar*


Agar is a gelatinous vegan substance derived from certain types of seaweed, making a stable gel structure when cooled down after being boiled. An agar phantom can be created by mixing agar, water, and salt to produce the recommended weight percentages of 4% and 0.5% for agar and salt, respectively, as per Equation (Equation 4) [9]:(4)$$\left\{\begin{array}{c} \frac{A}{A+w+s}=4.0\%\\ \frac{s}{A+w+s}=0.5\% \end{array}\right.\;\Rightarrow \;\left\{\begin{array}{c} A=\frac{8}{191}w\\ s=\frac{1}{191}w \end{array}\right.$$
where *A* is the weight of agar, *s* the weight of salt, and *w* the weight of water.

The phantom was estimated to have a capacity of 700 mL (equivalent to 700 g of water). By substituting *w* for that value in the above equations, we obtain the necessary quantities of 30 g of agar (*Hoosier Hill Farm*, €10.20 for 115 g from Amazon) and 4 g of salt, rounded up to the nearest unit. The mixture was prepared as follows:Boil 700 mL of regular tap water (or deionized water)Add 30 g of agar slowly while stirring the mixtureAdd 4 g of table salt while stirring until no granules are present (keep mixing while letting it cool down at room temperature for 10 min)Pour the mix into the assembled phantom through the top vents until the liquid reaches half the vent’s height and let it sit in a refrigerator until it fully solidifies (minimum 2 h, preferably overnight)

The resulting agar-based phantom is shown in Figure 21, left.


*Ballistic Gelatin (BG)*


BG is a common material used for making EEG phantoms, typically consisting of cattle/pork gelatin. Various weight percentage combinations of both gelatin and NaCl are shown to be an appropriate conductive medium [16]. Due to the limited accessibility of gelatin bulk powder (*YouHerbIt* 240 Bloom, €20.00 for 350 g of powder from Amazon), values within the middle range of the above were used, with a 15% weight percentage of gelatin and 8% weight percentage of NaCl, which yield Equation (Equation 5):(5)$$\left\{\begin{array}{c} \frac{BG}{BG+w+s}=15.0\%\\ \frac{s}{BG+w+s}=8\% \end{array}\right.\;\Rightarrow \;\left\{\begin{array}{c} BG=\frac{15}{77}w\\ s=\frac{8}{77}w \end{array}\right.$$

The same steps as for agar were followed to prepare the gelatin solution, with 140 g of gelatin powder mixed with 70 g of salt in 700 mL of hot (not boiling) water. Unlike agar, gelatin is better mixed at a lower temperature to prevent the formation of air pockets.

The resulting BG-based phantom is shown in Figure 21, right.


*Carbon Fiber-Doped Silicone (CF)*


A non-perishable phantom that can maintain signal integrity indefinitely was also envisioned, as the aforementioned organic materials tend to deteriorate over time.

Conductive fillers have been tested to make silicone rubbers conductive, with carbon fibers found to be the most effective even at a 0.5% weight percentage measuring a 2 kΩ resistance. Increasing to 1% further lowered the resistance to 200 Ω, and higher percentages gave similar resistances, indicating that the percolation threshold (filler concentration at which a conductive network is fully established) was reached [29].

A step-by-step approach for doping the silicone was followed [30], which used the following materials:Chopped carbon fibers, 3 mm in length, €30.00 for 500 g from AmazonTwo-part A/B system platinum-curable silicone, mixing ratio of 1:1, €23.00 for 630 mL from Amazon (two were required for the phantom)

The silicone was estimated to have a density of 1.11 g/cm^3^. Various carbon fiber weight percentages were tested according to Equation (Equation 6) to establish a percolation threshold:(6)$$\frac{CF}{CF+S}=X\%\;\left(where\;X=0.5,1.0,1.5,2.0\;...\right)$$
where *CF* is the weight of carbon fibers and *S* is the weight of silicone. It was observed that when the weight percentage of CF exceeded 1.0%, the mixture became too dense and unsuitable for molding with the current mixing methods. Therefore, only CF weight percentages of 0.5% and 1.0% were considered and prepared according to Equation (Equation 7):(7)$$\left\{\begin{array}{c} CF=\frac{S}{199}\;,\;for\;X=0.5\%\\ CF=\frac{S}{99}\;,\;for\;X=1.0\% \end{array}\right.\;$$

This non-perishable phantom was ultimately made for a CF weight percentage of 1.0%, with 8 g of carbon fibers (rounded to nearest unit) being used for a total of 700 mL of silicone, following the next steps [30]:Measure 8 g of carbon fibers into a disposable cup (use a mask and gloves when handling carbon fibers)Wet the carbon fibers with a small amount of rubbing alcohol, spread them around, and let it almost entirely evaporate (to release strands of hair that surround the carbon fibers)Add the carbon fibers to 350 mL of part A silicone and mix thoroughly until the mix presents a grey/blueish tint (an electric mixer with a wider spatula attachment was used)Add 350 mL of part B silicone and keep mixing for up to 25 min until it reaches the same tintPour into the phantom casing equally through each vent and let cure for 6 h.

To characterize the conductivity profiles of our chosen materials, fine strips of each material (30 × 10 × 2 mm) were prepared specifically for conductivity testing. Both ends of each sample were coated with conductive silver ink and the standard two-wire method was employed to measure the resistance across the sample. The conductivity of each sample was then calculated according to Equation (Equation 8):(8)$$\left\{\begin{array}{c} \rho =R\frac{A}{L}\\ \sigma =\frac{1}{\rho } \end{array}\right.\;\Rightarrow \;\sigma =\frac{L}{R\times A}$$
where ρ is the material resistivity, *R* is the measured resistance, *A* is the area of the cross-section of the sample, and *L* is the sample’s length. σ is the material’s conductivity as the inverse of resistivity. The measured conductivity values for each material are listed below in Table 1.

While the sample preparation for the CF-doped silicone samples was successful, the mixing method employed was not suitable for a homogeneous distribution of carbon fibers in the larger phantom mold. This resulted in the creation of conductive (darker) and non-conductive (translucent, low to no carbon fibers) areas, as illustrated in Figure 22. Therefore this material was omitted from further testing.

#### 2.2.2. Ear-EEG Phantom Testing Protocol and Setup

The agar and BG phantoms were tested over eight days, alternating daily for a total of four sessions each (i.e., “Day 4” is the last testing day per phantom), and refrigerated while not in use (4 °C). All phantoms were made using ear scans from the same subject such that the same pair of EEG earbuds could be used across all tests.

The phantom was first weighed without the side lids and with the AUX cables stacked on top. The antennas were connected to a signal generator, simulating a 10 Hz (alpha band) 100 mV square wave. The oscilloscope probe was jammed in the side of the phantom to measure the signal through the material alone and test for the material’s integrity. Then, both earpieces were inserted, and electrode contact impedances were measured using an analog impedance meter (D175, Digitimer), capable of measurement values up to 50 kΩ. The impedance meter’s reference electrode was connected to electrode Ex1. The oscilloscope probe was then connected directly to each electrode to visualize the simulated signal, which allowed us to test if any electrode was not functioning or had a diminished signal amplitude compared to the one measured through the material. To simulate an ear-EEG acquisition, the earbuds were then connected to an OpenBCI Cyton amplifier (8 channels, 250 Hz sampling rate, 24-bit ADC) and connected as per the recommended setup [31]. Ex1 was assigned to the bottom SRB2 pin (reference), and Ex2 was assigned to the bottom BIAS pin (noise-canceling) a priori. The remaining electrodes were connected in ascending order to the bottom N1P to N6P analog input pins (Ex3 is N1P and Ex8 is N6P on the board). Impedance values were also rechecked through the OpenBCI GUI. Then, 20 s recordings were obtained through the amplifier on each ear, with a gain of 24: a recording with the simulated wave and an acquisition with no signal to assess the recorded noise floor at each electrode, both offline bandpass-filtered between 0.3 and 100 Hz.

The protocol was repeated after applying conductive paste (Ten20) to each ear electrode. The testing setup is illustrated in Figure 23.

The RMS noise floor was calculated and the alpha simulation SNR was calculated (with 1% confidence) as per Equation (Equation 2).

#### 2.2.3. Phantom Integrity and Durability

When measuring the signal through the phantom, it is important to track the usability of this tool over time. This was achieved by monitoring the net weight of each phantom and measuring the input signal through the medium over several days.

Table 2 shows that by Day 4, the agar and BG phantoms lost no more than 10 g in weight (with a higher percentage loss in the agar). The reason for this weight loss can be attributed to the evaporation of water in both materials.

Table 3 shows the measures of signal integrity for each phantom over a one-week period. The agar phantom suffered the most significant amplitude loss, decreasing from an initial amplitude of 100 mV to an amplitude of 36 mV by day 4. The BG phantom signal reduced by 50% of its initial amplitude of 80 mV to 40 mV due to the drying of the material. Note, the phantoms endured extensive pilot testing before the first day of measurements and thus, were kept in sub-optimal refrigerating conditions, leading to the observed drying of the materials.

#### 2.2.4. Electrode Impedance

Figure 24 shows an example of electrode contact impedances (measured via the OpenBCI Cyton board and validated via the analog impedance meter) for wet- and dry-electrode conditions taken during the agar phantom’s second day of recordings. Impedance measurements show a clear improvement across electrodes when applying conductive paste to the electrodes, reliably bringing the impedance under 10 kΩ, which is an acceptable level for EEG recordings (the same effect happened for BG measurements, with overall lower values than agar). From these data, we can see that channels ER3, EL3, EL7, and EL8 did not achieve suitable impedances in the dry-electrode condition, with values greater than 50 kΩ, indicative of no or very poor electrode contact. This effect was consistent across testing sessions (suggesting ER4 as a better channel than ER3 for dry recordings, at least on this specific subject). However, applying conductive paste to these electrodes reduced the impedances to a suitable level.

#### 2.2.5. Noise Floor Measurements

As with the impedance measurements, noise floor measurements (without the simulated input signal) greatly benefited from the use of conductive paste on each electrode. As seen in Figure 25, applying conductive paste to the electrodes lowered the noise floor close to or under 1 µVrms across all channels. These observations indicate that the electrode material and shape are adequate for wet-electrode recordings. Comparing the left- and right-ear results in the dry condition, we would predict that the left ear is more prone to noise (with RMS values on the order of hundreds of µV) when compared to the right-ear electrodes, suggesting better performance in the right earpiece of this particular subject.

#### 2.2.6. Alpha-Wave Simulation

When simulating a known signal into the phantom through the two antennas, the 10 Hz input signal (to simulate alpha-band activity) was successfully recorded at all ear electrode sites on Day 1 of testing in agar and BG. As expected, the modulation SNR was higher in the ear canal electrodes (e.g., ER8), as they are further away from the within-ear reference (ER1) than, for example, ER3 or ER4.

In Figure 26, the resulting alpha modulation spectra at ER8 are present, with clear peaks at the 10 Hz and 30 Hz odd harmonic frequencies that compose the square wave. The noise floor of the spectra resembles what would be expected from an EEG recording, although that was not always the case in the following days of testing when the agar or BG phantom noise floor readings started to increase. Nonetheless, the phantom antenna signal delivery and materials proved effective in simulating a given neural signal, and for this purpose, provide a feasible validation framework.

## 3. Results: Toolkit Use Case: Validation of an Ear-EEG Sensor

### 3.1. Ear-EEG Devices and Setup

The tested ear-EEG devices were developed by industry partner Segotia and fabricated by a third-party vendor (Figure 27). Each earbud was custom-fit to the participant’s ear (five subjects total, age avg. 36.2 ± 10.6 std.), based on a wax mold of their outer ear taken by a professional audiologist. This mold was scanned and digitized as an *.stl* file, digitally rendered into a discrete earbud form factor, and 3D-printed at a very high quality. On the external side, the earbud boasts an earphone insertion vent for sound delivery.

Each earbud contains eight 2 × 1 mm sintered Ag/AgCl disc electrodes by Easycap GmbH, Germany (Figure 27). One is located in the concha cymba area (1), four around the concha cavum (2–5), one near the antitragus (6), and two inside the ear canal (7 and 8) oriented in the posterior and anterior directions, respectively. Thin wires were welded to the electrodes, which exit the earbud near electrode 6. The electrode cabling was carefully color-coded for connecting to the amplifier (see Figure 27, left).

The ear-EEG data were acquired under wet- and dry-electrode conditions, simultaneously with conventional wet scalp-EEG using the same amplifier to provide a direct comparison with our scalp-EEG benchmark. However, since this amplifier only supports 24 recording channels outside the CMS and DRL, four scalp channels (F3, F4, P3, P4) from the control group were disconnected to accommodate all ear-EEG electrodes on the amplifier. The resulting ear- and scalp-EEG electrode configurations used are shown in Figure 28.

The EEG earbuds were strapped to the participant’s back and shoulders. The ear-EEG cabling was taped to the cap with skin-safe tape and connected to the amplifier, as seen in Figure 29.

A small amount of conductive electrode paste (Ten20) was used for wet-electrode ear recordings. Subjects inserted the earpieces themselves in a way that was snug and yet comfortable. Ear electrode impedance was generally too high or immeasurable by the Streamer software, and as such was not assessed. For the ear-EEG recordings, the lowest volume setting from the validation group was adopted for all subjects.

The same data preprocessing and statistical analysis were applied to the ear-EEG data as before. To identify the optimal ear-EEG re-referencing strategy, the following reference configurations were systematically tested:**Cz:** standard central scalp reference**T8:** scalp reference closer to the ear**ER3:** within-ear reference (e.g., relative to ER8) and between-ear reference (e.g., relative to EL8).

### 3.2. EaR-P Lab for Ear-EEG Validation

This section presents the ear-EEG data obtained for each of the EEG validation paradigms (using wet-electrode recordings), demonstrating how this tool can be used to assess novel ear-EEG devices.

#### 3.2.1. Alpha Block

Figure 30 shows the grand average alpha power at ER8/EL8 for different ear-EEG reference configurations. Alpha modulation was 3.7 dB for the Cz reference, approximately 3 dB lower than the scalp-EEG benchmark taken at Oz. Re-referencing to T8, closer to the ear, alpha modulation decreases to 3.1 dB. Using a within-ear and between-ear reference resulted in lower modulation values of around 2 dB.

#### 3.2.2. ASSR

The ASSR SNR at ER8 (Figure 31) for the Cz reference had an average value of 7.9 dB, a decrease of 1.5 dB relative to the scalp-EEG benchmark. Comparing the scalp and ear-EEG reference strategies, we see that the SNR is 1.2 dB higher for the within-ear ER3 reference over the temporal T8 site.

#### 3.2.3. SSVEP

The SSVEP measured at ER8 had an SNR of 6.4 dB and 5.9 dB when referenced to Cz and T8, respectively. This was much lower than the scalp-EEG benchmarks of 11 dB and 7.5 dB measured at Oz and T8, respectively. However, in-ear measures using the within-ear and between-ear referencing configurations were not significantly greater than zero, even for other in-ear electrodes (Figure 32).

#### 3.2.4. AEP

When referenced to Cz, the AEP response at electrode ER8 displayed a significant N1 component (albeit inverted) with an amplitude of around 6 µV for wet-electrode ear-EEG recordings. Referenced to T8, the N1 component was diminished but significant, with a peak amplitude of around 2 µV (Figure 33).

For the ear reference analysis (Figure 34), only the between-ear configuration measured at EL8 showed a significant N1 component with an amplitude of 1 µV.

#### 3.2.5. VEP

The VEP waveform measured at Oz for the scalp-EEG benchmark was not evident near the ear at T8. For ear-EEG data, the VEP waveform was visible when using both the Cz and T8 referencing configuration, with amplitudes of around 4 and 2 µV, respectively (Figure 35).

For the ER3 reference, only the between-ear configuration resulted in significant deflections in the VEP waveform, showing a 200 ms component with an amplitude of 1 µV (Figure 36).

#### 3.2.6. AEP OddBall (MMN)

Significant MMN responses were observed at ER8 for both the Cz and T8 referencing configurations, with the deviant response being more prominent in the former (Figure 37).

For within-ear and between-ear referencing, no significant components resembling an MMN response were found.

#### 3.2.7. VEP OddBall (P300)

The greatest observed difference between the target and standard VEP waveforms was a negative deflection at a latency of around 350 ms for Cz (5 µV) and T8 (2 µV) referencing, suggesting a later onset for this component than that observed on the scalp (Figure 38).

For within-ear and between-ear referencing, no significant components resembling a P300 response were found.

#### 3.2.8. EOG (Blinks and Saccades)

Examining EOG blink ratios, the ER3 reference resulted in a blink ratio about 1.5 times higher than the scalp references near the ear, with diminished results for the between-ear configuration at EL8. The best ratio was obtained for the within-ear reference, with a blink ratio of 3 (Figure 39).

Saccade profiles at ER8 were maintained for each of the different eye directions, with greater amplitudes observed for the Cz reference, but better differentiation when referencing to T8 with a 10 µV difference between up and down saccades (Figure 40).

For within-ear referencing, the vertical saccade plane presents a better differentiation of 3 to 4 µV, while referencing between ears resulted in a greater amplitude difference between right and left saccades (the vertical saccades present the same profile, Figure 41).

### 3.3. Reassessment of Dry Ear-EEG ASSR Data

The ear-EEG phantom results suggest that the ER4 electrode has better connectivity to the ear than ER3, which was previously selected as a reference, for the live ear-EEG recordings. Figure 42 shows a direct comparison of the ASSR dry data (most relevant auditory paradigm) reanalyzed with ER4 as a reference for this single subject that was used for molding the phantom. The suggested electrode ER4 from the phantom improved the SNR at ER8 (and the other electrodes) by about 2.3 dB, demonstrating how the phantom can be used to predict improved performance of ear-EEG devices without the need to conduct testing on a live subject.

## 4. Discussion

Ear-EEG devices are an active field of research and development with potential to disrupt the way in which brain activity can be factored into daily life. We can expect an increase in new applications, form factors, and sensor technology as the field grows. Currently, there is scientific evidence to support the use of ear-EEG [20,21] and we set out to provide a set of tools that could benefit further testing and characterization of ear-EEG devices. Our approach was to develop a validation toolkit that would enable the characterization of ear-EEG devices from hardware to neural signal acquisition.

We developed an EEG acquisition application based on nine commonly-used EEG paradigms: EaR-P Lab. Our validation results indicate that EaR-P Lab can indeed be used to elicit the expected neural responses from scalp-EEG recordings. When evaluating these responses from the perspective of near-ear electrode locations such as T8, we observed that VEP responses were the most affected, indicating that ear-EEG may have a limited capability for more visually-driven appellations.

We also developed an EEG phantom suitable for evaluating ear-EEG sensors by combining 3D scans of a participant’s ear impressions and 3D printing a mold to be filled with conductive material. This allowed us to assess ear-EEG devices in terms of their electrode contact impedance, noise floor, and acquisition of a known (synthetically generated) signal. To the best of our knowledge, this is the first EEG phantom dedicated for ear-EEG devices, as traditional head phantoms neglect the structures of the ear needed for this type of detailed evaluation [7].

Our phantom development highlighted salt-doped agar as the best substrate material for this purpose, with conductivity values equivalent to those reported in the literature for whole-brain anatomy at 0.33 S/m [32,33]. While salt-doped BG and CF-doped silicone were also investigated as potential material substrates, there were notable limitations with these options.

BG showed a drastic decrease in signal transmission integrity across a week of testing. This could be related to the storage conditions, highlighting the susceptibility of the phantom to environmental conditions over time, which could hinder the repeatability of results when evaluating ear sensors. While agar is also an organic and perishable material, it showed a more stable performance despite being kept under the same storage conditions as the BG phantom. To address the susceptibility to degradation over time, we investigated the use a synthetic non-perishable material composition for the phantom by doping platinum-cured silicone with carbon fibers. This approach, however, did not scale from the prepared samples for conductivity testing to the larger ear-EEG phantom.

The mechanism by which carbon fibers turn silicone into a conductive medium is different to that of salt-doped agar or BG. In the latter, salt fully dissolves in the medium, creating a homogeneous conductive substrate, while in the former, carbon fibers disperse in the silicone, creating conductive paths through heterogeneously mixed fibers. When the samples were small strips, this dispersion of carbon fibers was effective throughout the volume; however, when a larger volume was used, the stirring method did not achieve dispersion of fibers over the full volume. This led to patches of non-conductive silicone, clearly visible in the final product. A more optimal mixing approach should be devised in the future to ensure that the CF phantom is conductive throughout. It should be noted, however, that the conductivity of this synthetic substrate is orders of magnitude larger than that of real anatomical structures and the agar/BG substrates. As an alternative to silicone, other commercially available synthetic ballistic gels could be considered (e.g., Perma-Gel, Inc. or Humimic Medical, TM). However, as these are based on a combination of mineral oil and proprietary gellants, new methods to induce and control conductivity would need to be developed.

We validated these tools using custom-fit ear-EEG devices developed in-house at Segotia. Our results showed that the ear-EEG sensors were functional and EEG responses were successfully recorded when using a scalp reference (Cz and T8). However, in-ear and between-ear referencing configurations only showed significant EEG responses for steady-state paradigms and alpha blocking (although diminished AEP and VEP responses could also be seen). Further investigation is needed to identify the optimal referencing configuration for each of these ERP paradigms, as it has been shown that this is crucial for obtaining characteristic ERP responses [34]. However, this was not the scope of the current study. Importantly, we could see clear EOG deflections using both within-ear and between-ear reference configurations, which has promising implications for ear-based BCI technologies.

We also demonstrated how the data obtained with our ear-EEG phantom—without the need of a testing subject—could be used to select an optimal referencing location in the ear and when applied to the actual ear-EEG recordings of the same tested subject, it improved the SNR of the ASSR response.

Future work for our ear-EEG validation toolkit include improving the testing protocol and the assessment of a generic ear canal for comparing different earpiece designs. Our ear-EEG phantom methodology could also be improved in the future for studying how real-world factors, such as gait, impact ear-EEG [35].

The presented ear-EEG validation toolkit is available to the scientific community via the Open Science Framework (OSF) repository named “Brain Wearables: Validation toolkit for Ear Level EEG sensors” (https://osf.io/2dxs4/). All future modifications to the Ear-P Lab and the ear-EEG phantom developed by the authors will also be updated in the referenced repository.

## 5. Conclusions

In this study, we set out to address the need for a comprehensive characterization tool for ear-EEG sensors at the neural and hardware level. We presented a validation toolkit composed of a desktop application (EaR-P Lab) for acquiring EEG data to nine cardinal EEG paradigms relevant to the field of ear-EEG and a novel ear-EEG phantom for interrogation of the sensor and hardware signal chain quality. To facilitate better characterization of ear-EEG devices, our ear-EEG phantom is the first electrical phantom (designed based on 3D impressions of the human ear) to include the ear canal structures. These tools were validated with traditional scalp recordings and bench-top evaluations before being applied in a use-case study, which characterized the functionality of a custom-made ear-EEG device.

Through our use-case study, we demonstrated the utility of this toolkit for the characterization of ear-EEG sensors and relating hardware level observations, such as sensor placement and contact, to neural level outcomes, such as ASSR signal strength. To our knowledge, this is the first publication where controlled testing through an EEG phantom is applied to ear-EEG technology. By making this toolkit available through the Open Science Framework, we hope to facilitate and accelerate future research on and development of ear-EEG sensors and devices.

## Figures and Tables

**Figure 1 sensors-24-01226-f001:**
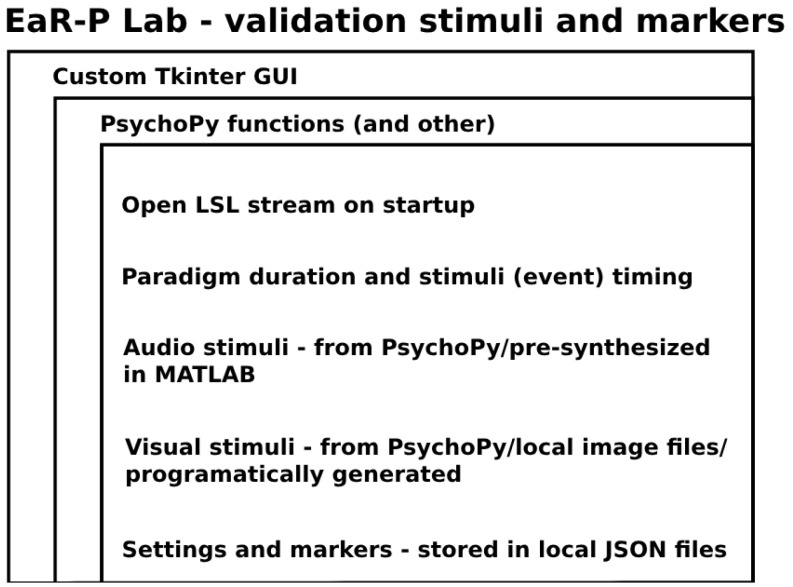
**EaR-P Lab**—structure and main attributes.

**Figure 2 sensors-24-01226-f002:**
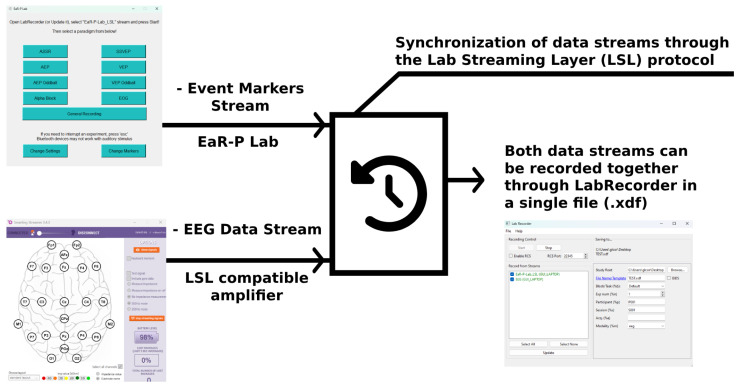
**EaR-P Lab**—schematic of functional framework.

**Figure 3 sensors-24-01226-f003:**
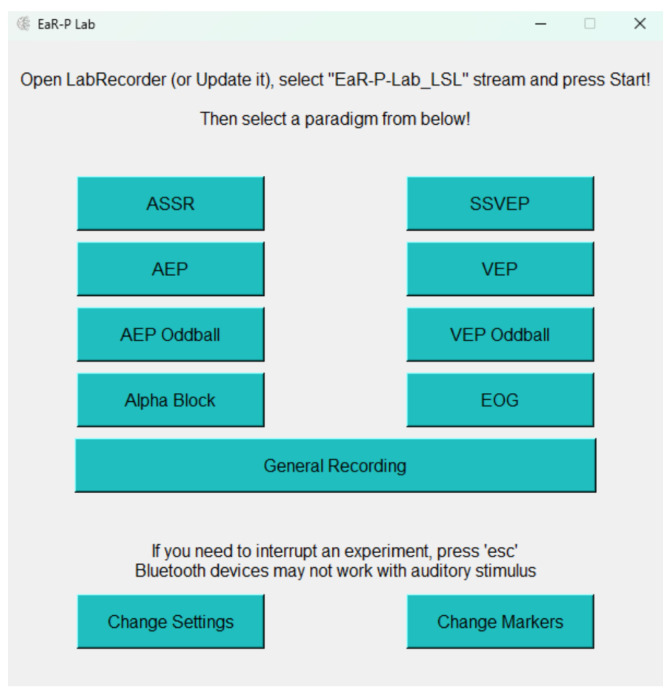
**EaR-P Lab**—main menu.

**Figure 4 sensors-24-01226-f004:**
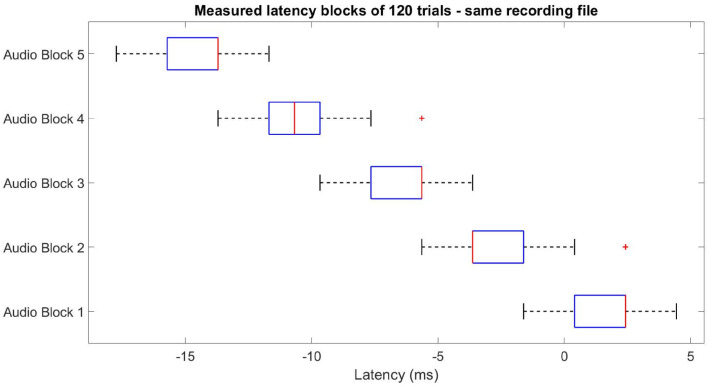
Latency variation when recording multiple event-related potential (ERP) blocks on the same file, exemplified for auditory stimuli—a similar effect happens for visual stimuli.

**Figure 5 sensors-24-01226-f005:**
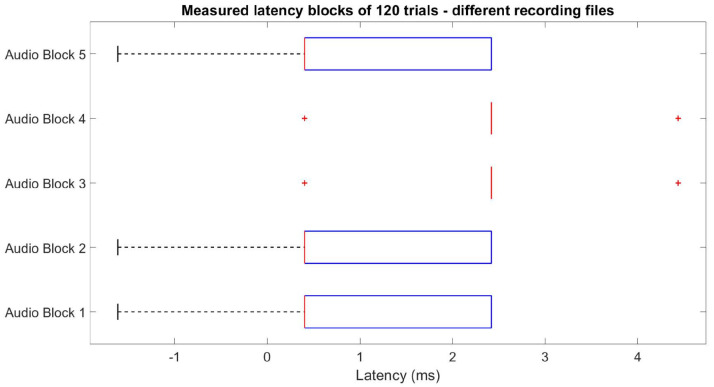
Nullified cascading effect is when recording multiple ERP blocks in different files after restarting data streaming, exemplified for auditory stimuli—a similar effect happens for visual stimuli.

**Figure 6 sensors-24-01226-f006:**
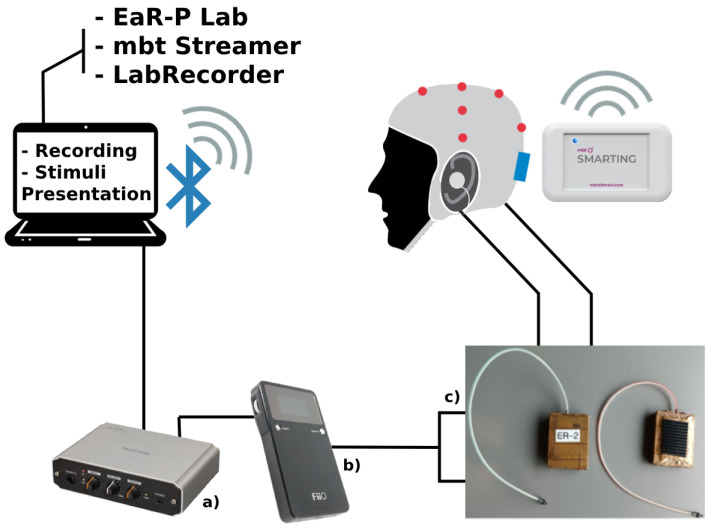
EEG acquisition setup schematic and equipment: (a) USB audio interface TASCAM US-100; (b) digital-analog converter (DAC) amplifier FiiO Alpen 2; (c) ER2 etymotic tubal-insert research-grade earphones.

**Figure 7 sensors-24-01226-f007:**
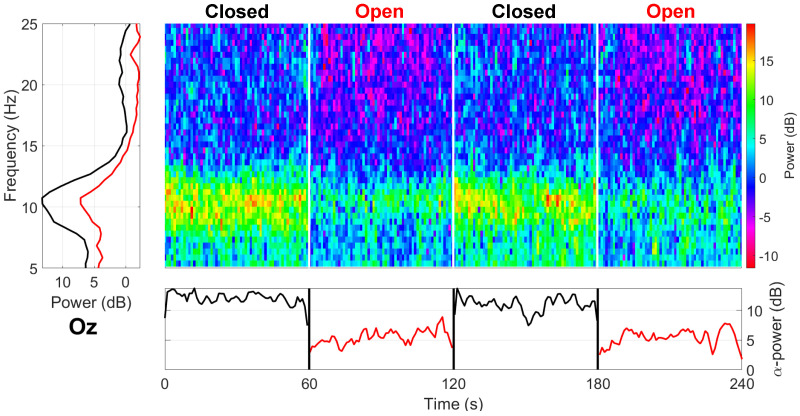
Grand average spectrogram for the alpha block paradigm at Oz (Cz referenced). The bottom horizontal plot shows the mean alpha power (8 Hz) as a function of eye state, while the left vertical plot shows the frequency response for the two conditions.

**Figure 8 sensors-24-01226-f008:**
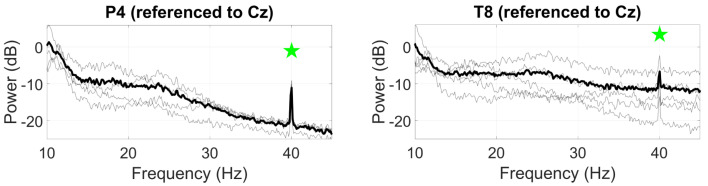
Grand average ASSR responses (black line) to a 40 Hz AM auditory stimulus at P4 (**left**) and T8 (**right**). Statistically significant peaks are highlighted by the green star token, based on an *F*-test (p<0.05), grey lines represent individual responses.

**Figure 9 sensors-24-01226-f009:**
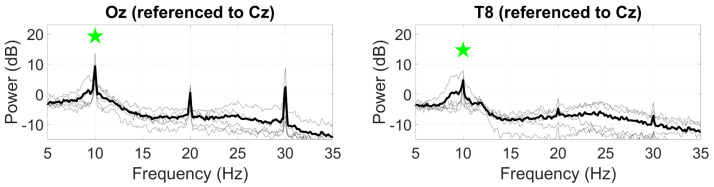
Grand average SSVEP responses (black line) to a 10 Hz visual stimuli at Oz (**left**) and T8 (**right**). Statistically significant peaks are highlighted by the green star token, based on an *F*-test (p<0.05), grey lines represent individual responses. Only the first harmonic was statistically evaluated.

**Figure 10 sensors-24-01226-f010:**
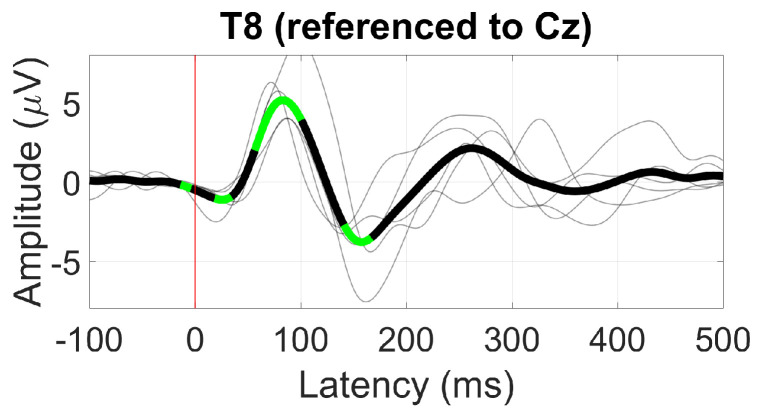
Grand average AEP waveform (black line) at T8. Statistically significant segments are highlighted in green, based on *t*-tests (p<0.05, not corrected for multiple comparisons), grey lines represent individual responses.

**Figure 11 sensors-24-01226-f011:**
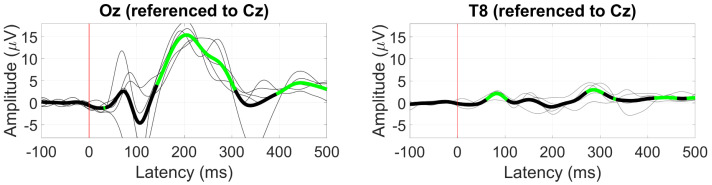
Grand average VEP waveform (black line) at Oz (**left**) and T8 (**right**). Statistically significant segments are highlighted in green, based on *t*-tests (p<0.05, not corrected for multiple comparisons), grey lines represent individual responses.

**Figure 12 sensors-24-01226-f012:**
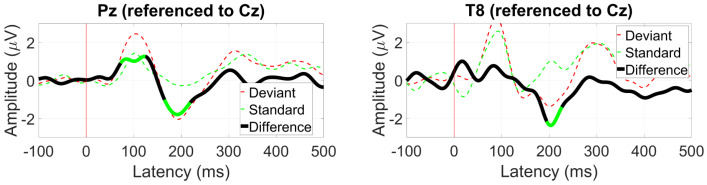
Grand average mismatch negativity (MMN) waveform at (**left**) Pz and (**right**) T8. Statistically significant segments are highlighted in green, based on a *t*-test (p<0.05, not corrected for multiple comparisons).

**Figure 13 sensors-24-01226-f013:**
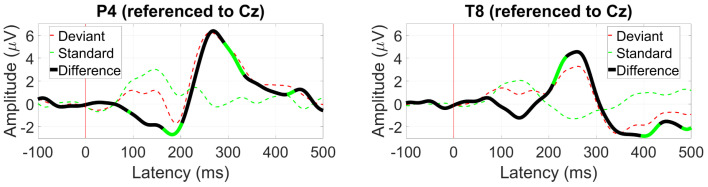
Grand average P300 waveform at (**left**) P4 and (**right**) T8. Statistically significant segments are highlighted in green, based on a *t*-test (p<0.05, not corrected for multiple comparisons).

**Figure 14 sensors-24-01226-f014:**
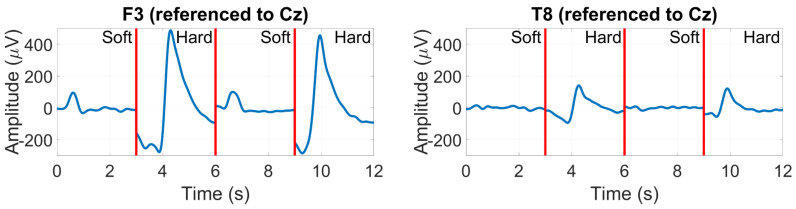
EOG amplitudes for soft and hard blinks in an example subject recorded at F3 (**left**) and T8 (**right**).

**Figure 15 sensors-24-01226-f015:**
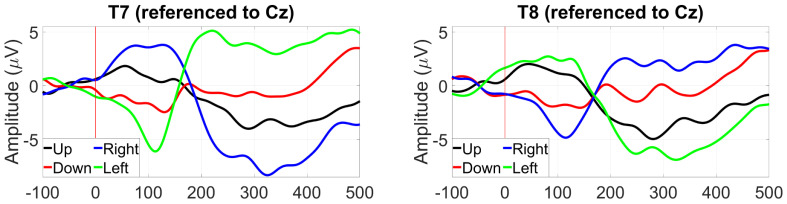
Grand average saccade profiles in the four cardinal directions, at T7 (**left**) and T8 (**right**).

**Figure 16 sensors-24-01226-f016:**
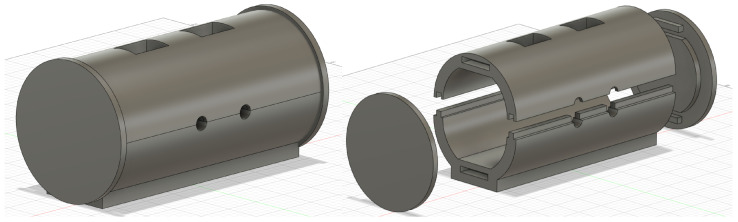
CAD drawings of ear-EEG phantom mold casing. Closed render of the mold (**left**). Exploded render of the mold (**right**).

**Figure 17 sensors-24-01226-f017:**
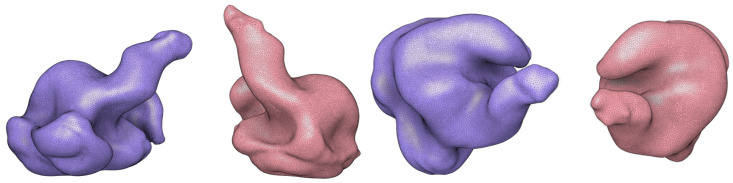
Outer ear scans (blue: left ear, red: right ear) from an example subject shown in elevation (**left**) and plan (**right**) view. The scans were obtained by an expert audiologist and digitized as *.stl* files.

**Figure 18 sensors-24-01226-f018:**
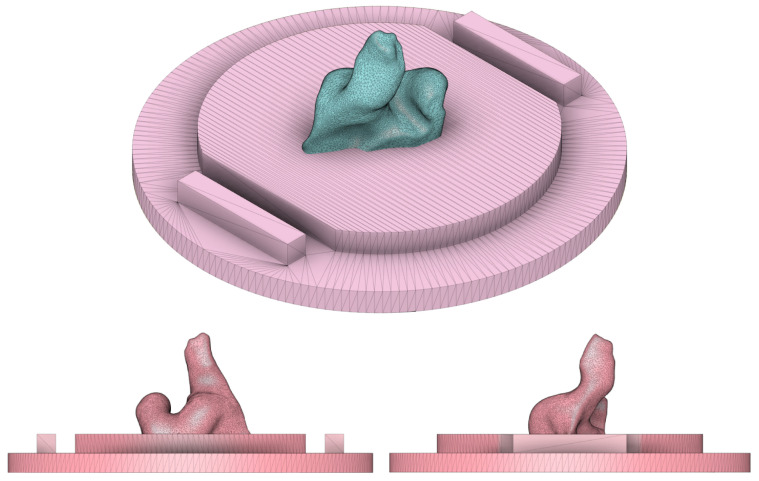
Example of a left ear scan being centered and oriented with the phantom’s lid mesh. Different views of the alignment and depth of the ear mesh and the lid mesh into a single rendered object are shown below.

**Figure 19 sensors-24-01226-f019:**
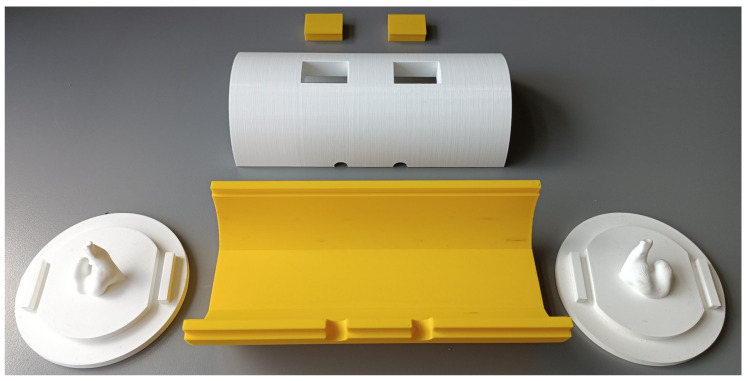
Disassembled ear-EEG phantom: bottom half (yellow), top half (white), and two lids with a left and right ear imprint from one of the test subjects.

**Figure 20 sensors-24-01226-f020:**
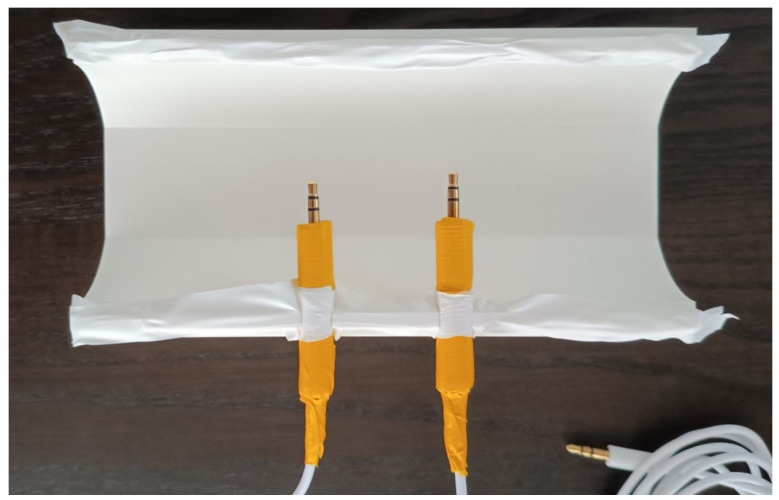
Ear-EEG phantom assembly—antennas and railing fittings were sealed with tape.

**Figure 21 sensors-24-01226-f021:**
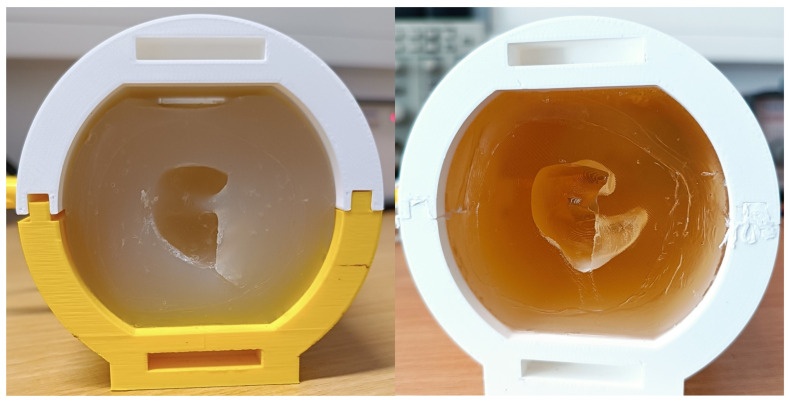
Ear-EEG phantoms made with agar (**left**) and ballistic gelatin (BG) (**right**).

**Figure 22 sensors-24-01226-f022:**
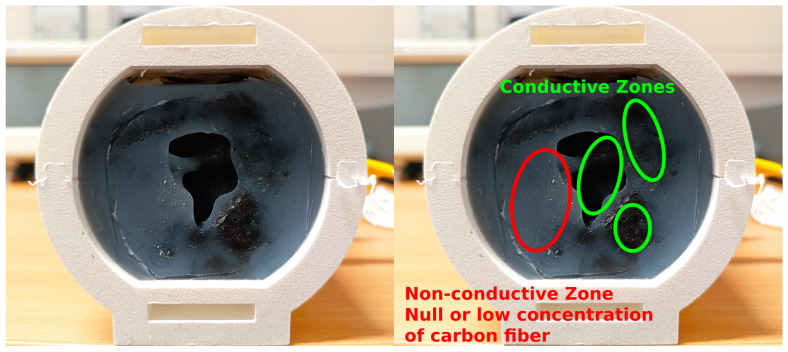
CF-doped silicone ear-EEG phantom—the lack of conductive homogeneity is highlighted on the right, with conductive and non-conductive zones visible.

**Figure 23 sensors-24-01226-f023:**
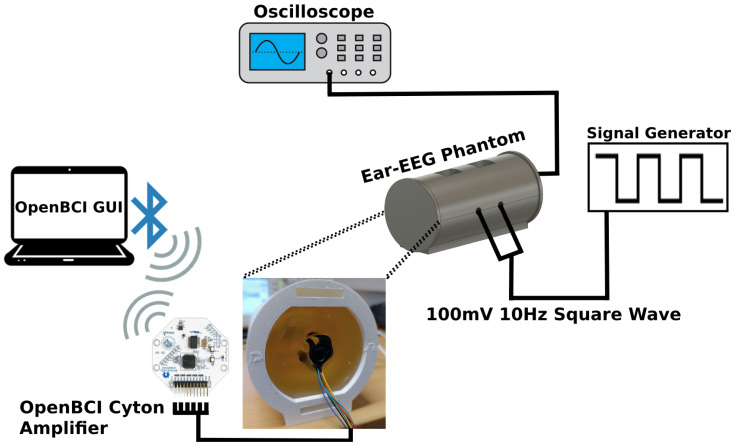
Schematic of testing setup of the proposed ear-EEG phantom.

**Figure 24 sensors-24-01226-f024:**
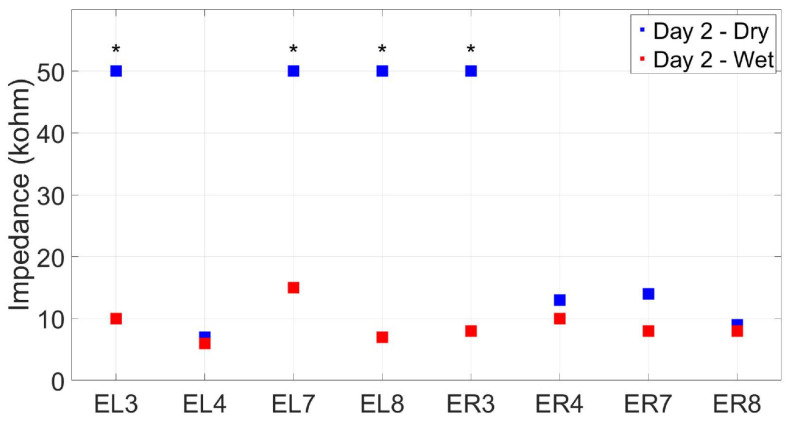
Contact impedance measures (kΩ) for the agar ear-EEG phantom for wet- and dry-electrode conditions (taken on Day 2 of testing). * indicates electrodes that surpassed an impedance of 50 kΩ in the dry condition.

**Figure 25 sensors-24-01226-f025:**
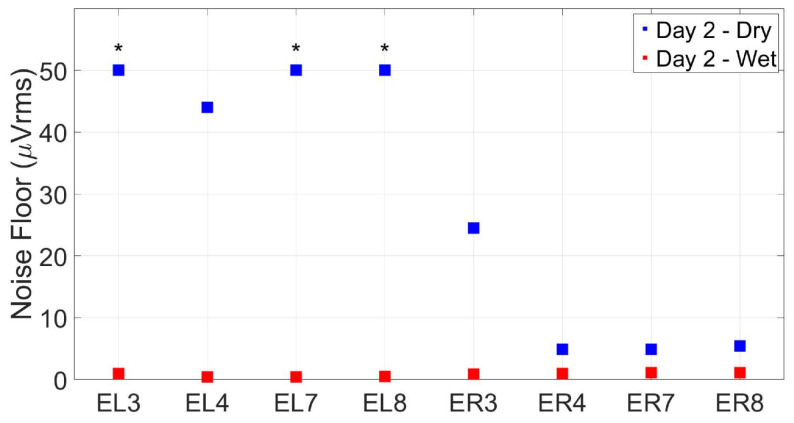
Noise floor measures (µVrms) for the agar ear-EEG phantom for wet- and dry-electrode conditions (taken on Day 2 of testing). * indicates electrodes that surpassed a noise floor of 50 µVrms in the dry condition.

**Figure 26 sensors-24-01226-f026:**
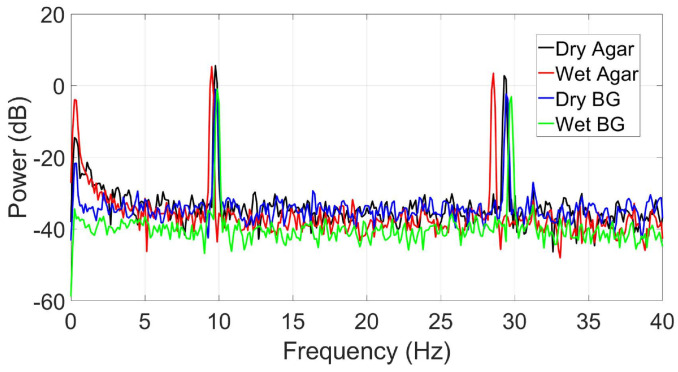
Power spectrum (dB) showing the synthetically generated alpha wave (10 Hz input signal) recorded using a custom ear-EEG device (electrode ER8) for the agar and BG phantoms in dry-and wet-electrode conditions.

**Figure 27 sensors-24-01226-f027:**
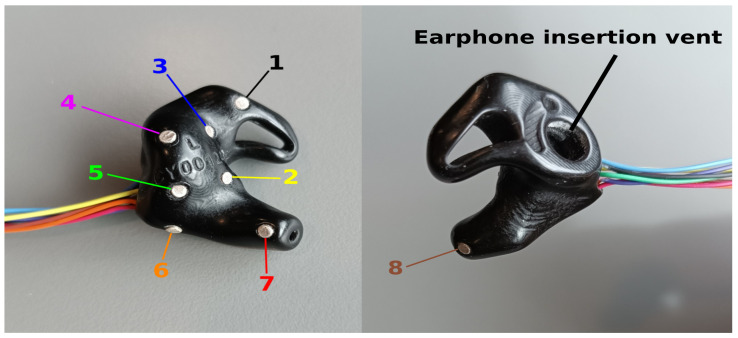
EEG earbuds developed by Segotia. Internal side of the tested earbuds (**left**). External side of the tested earbuds (**right**). Senors are numbered in order of signal channel acquisition 1–8.

**Figure 28 sensors-24-01226-f028:**
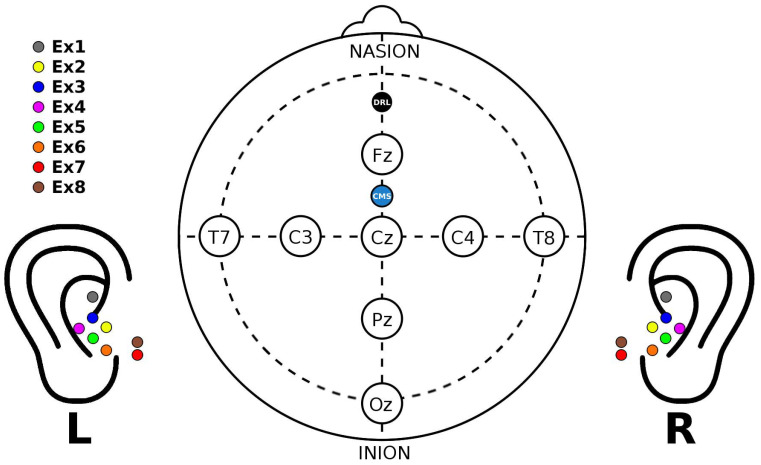
Ear- and scalp-EEG electrode configurations color-coded as in Figure 27. Electrode numbers are provided in the legend, where “**x**” is replaced by **L** or **R** to indicate the left or right ear, respectively.

**Figure 29 sensors-24-01226-f029:**
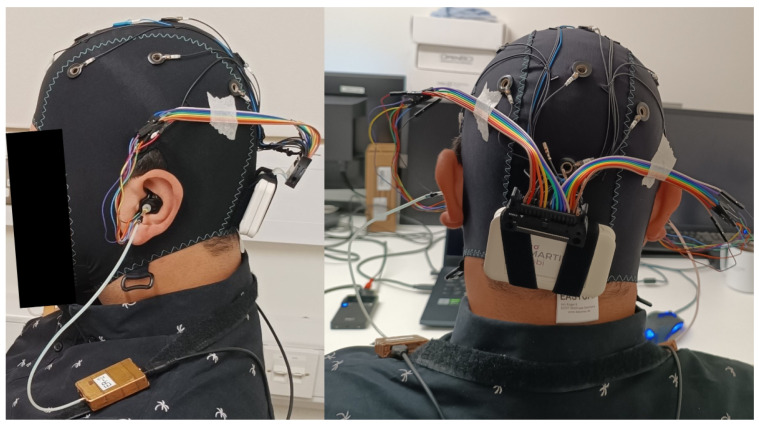
Ear- and scalp-EEG setup for side view (**left**) and posterior view (**right**).

**Figure 30 sensors-24-01226-f030:**
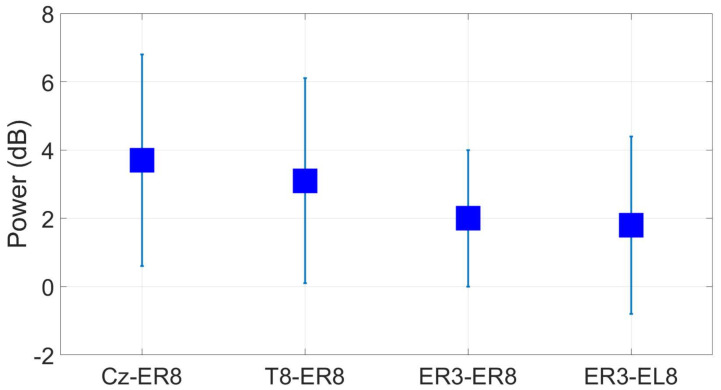
Grand average alpha modulation (wet ear-EEG) at ER8/EL8 for different referencing configurations. Omitted results are not significant based on a *t*-test (p<0.05).

**Figure 31 sensors-24-01226-f031:**
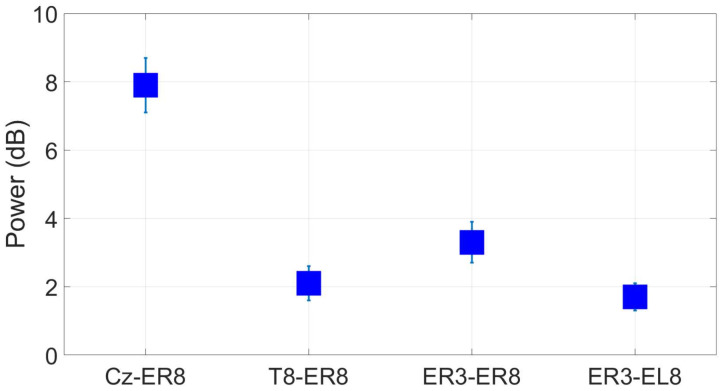
Grand average ASSR responses (wet ear-EEG) to a 40 Hz AM auditory stimulus at ER8/EL8 for different referencing configurations. Omitted results are not significant based on an *F*-test (p<0.05).

**Figure 32 sensors-24-01226-f032:**
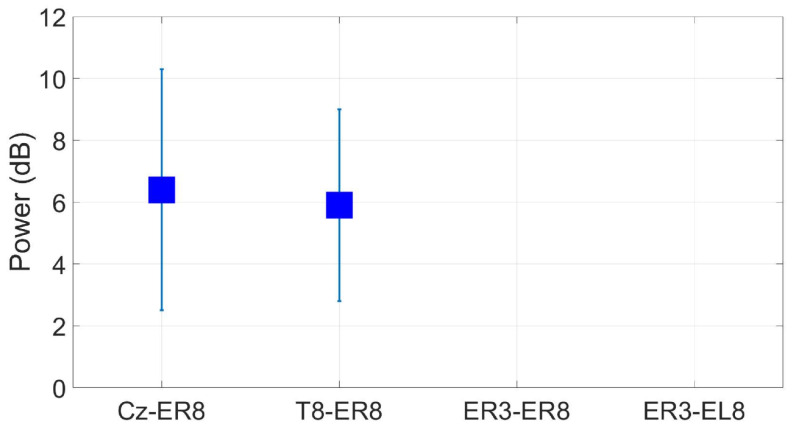
Grand average SSVEP responses (wet ear-EEG) to a 10 Hz visual stimulus at ER8/EL8 for different referencing configurations. Omitted results are not significant based on an *F*-test (p<0.05).

**Figure 33 sensors-24-01226-f033:**
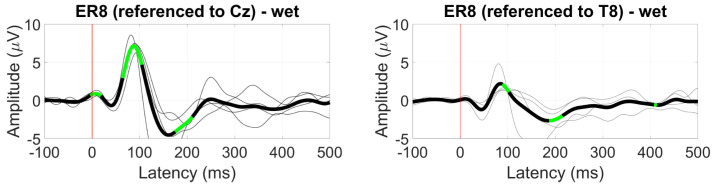
Grand average AEP waveform (black line, wet ear-EEG) at ER8 referenced to Cz (**left**) and T8 (**right**). Statistically significant segments are highlighted in green, based on a *t*-test (p<0.05, not corrected for multiple comparisons). Grey lines represent individual responses.

**Figure 34 sensors-24-01226-f034:**
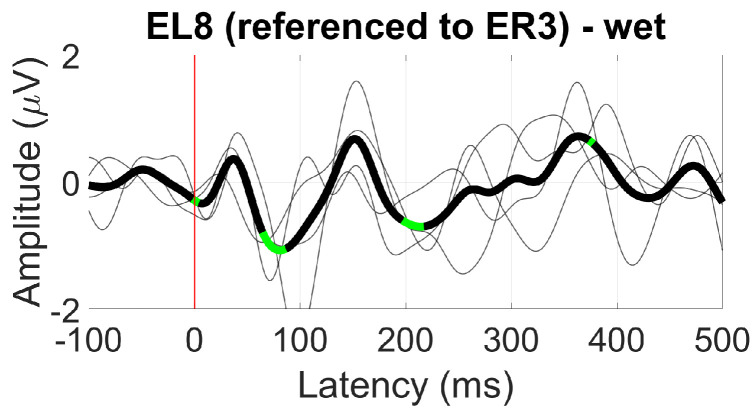
Grand average AEP waveform (black line, wet ear-EEG) at EL8 referenced to ER3. Statistically significant segments are highlighted in green, based on a *t*-test (p<0.05, not corrected for multiple comparisons). Grey lines represent individual responses.

**Figure 35 sensors-24-01226-f035:**
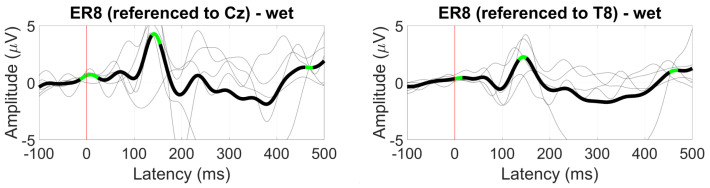
Grand average VEP waveform (black line, wet ear-EEG) at ER8 referenced to Cz (**left**) and T8 (**right**). Statistically significant segments are highlighted in green, based on a *t*-test (p<0.05, not corrected for multiple comparisons). Grey lines represent individual responses.

**Figure 36 sensors-24-01226-f036:**
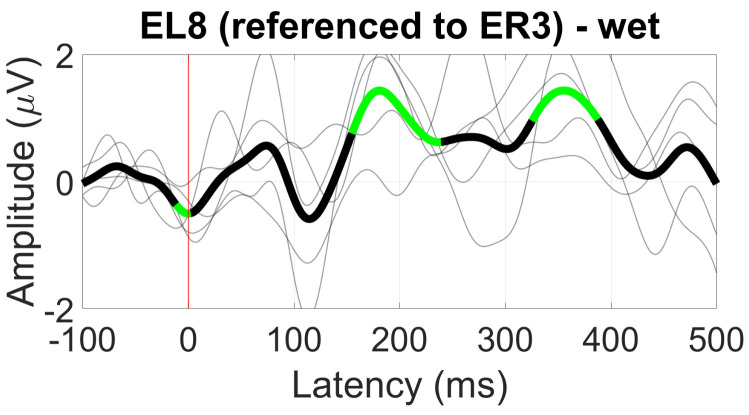
Grand average VEP waveform (black line, wet ear-EEG) at EL8 referenced to ER3. Statistically significant segments are highlighted in green, based on a *t*-test (p<0.05, not corrected for multiple comparisons). Grey lines represent individual responses.

**Figure 37 sensors-24-01226-f037:**
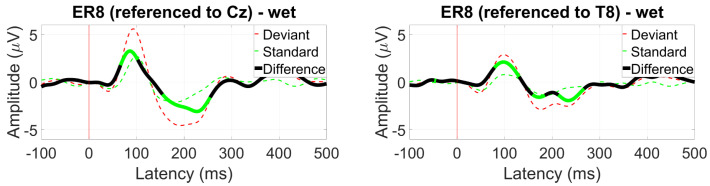
Grand average MMN waveform (wet ear-EEG) at ER8 referenced to Cz (**left**) and T8 (**right**). Statistically significant segments are highlighted in green, based on a *t*-test (p<0.05, not corrected for multiple comparisons).

**Figure 38 sensors-24-01226-f038:**
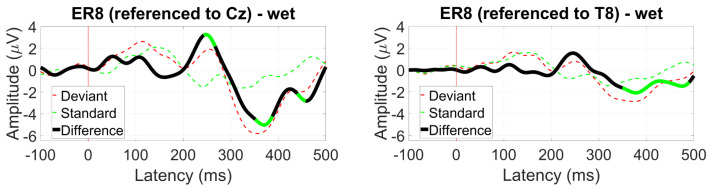
Grand average P300 waveform (wet ear-EEG) at ER8 referenced to Cz (**left**) and T8 (**right**). Statistically significant segments are highlighted in green, based on a *t*-test (p<0.05, not corrected for multiple comparisons).

**Figure 39 sensors-24-01226-f039:**
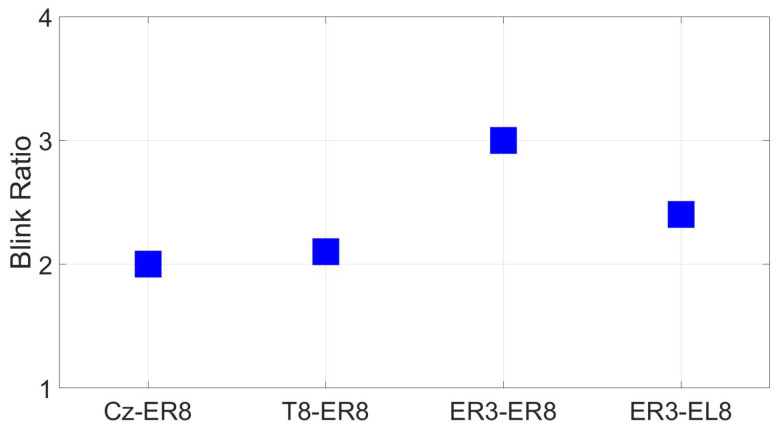
EOG amplitude ratio for soft and hard blinks (wet ear-EEG) for different reference configurations.

**Figure 40 sensors-24-01226-f040:**
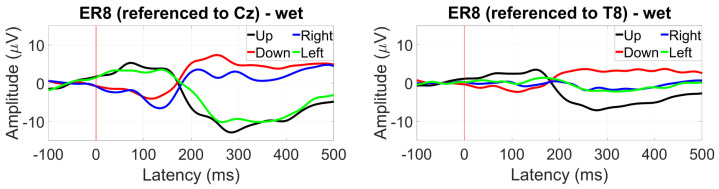
Grand average saccade profiles (wet ear-EEG) in the four cardinal directions at ER8 referenced to Cz (**left**) and T8 (**right**).

**Figure 41 sensors-24-01226-f041:**
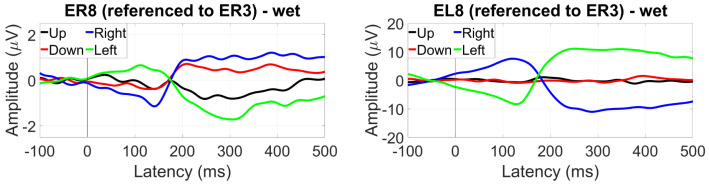
Grand average saccade profiles (wet ear-EEG) in the four cardinal directions at ER8 referenced within-ear (**left**) and between ears (**right**).

**Figure 42 sensors-24-01226-f042:**
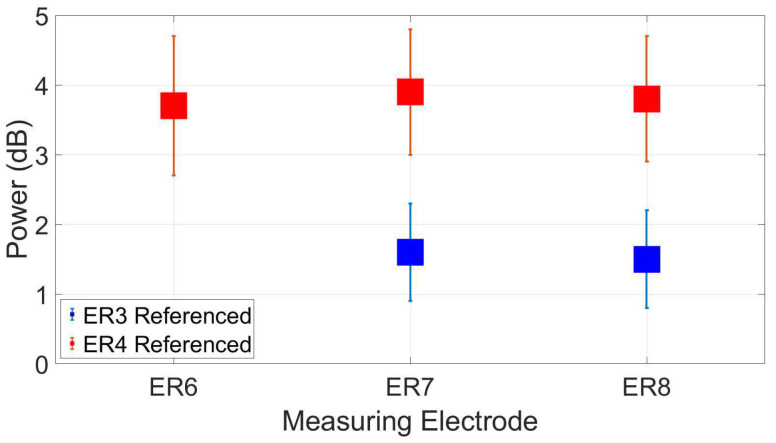
Reassessment of dry-electrode ear-EEG ASSR data for the subject used in the construction of the ear-EEG phantom. Data were re-referenced from ER3 (original reference) to ER4 (better electrode proposed by the phantom), resulting in an increase in SNR.

**Table 1 sensors-24-01226-t001:** Electrical conductivity (Siemens per meter) for samples of agar, BG, and CF-doped silicone (1%) as the proposed materials for the ear-EEG phantom.

	Conductivity [S/m]
**Agar**	0.309
**BG**	0.918
**CF (1%)**	14.035

**Table 2 sensors-24-01226-t002:** Measured mass (g) of the agar and BG ear-EEG phantoms over the testing days.

	Day 1	Day 2	Day 3	Day 4
**Agar**	855	851	850	845
**BG**	963	959	958	956

**Table 3 sensors-24-01226-t003:** Measured signal amplitude (mV) at the sides of each ear-EEG phantom directly through the material as a measure of signal integrity on agar and BG.

	Day 1	Day 2	Day 3	Day 4
**Agar**	44	40	40	36
**BG**	80	52	52	40

## Data Availability

The presented ear-EEG validation toolkit and the validation data are available to the scientific community via the Open Science Framework (OSF) repository named: “Brain Wearables: Validation toolkit for Ear Level EEG sensors” (https://osf.io/2dxs4/). All future modifications to the Ear-P Lab and the ear-EEG phantom generated by the authors will also be updated in the mentioned repository.

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
