# Peer review of "Brain Wearables: Validation Toolkit for Ear-Level EEG Sensors"

_sensors, 2024, doi:10.3390/s24041226_

Round 1

Reviewer 1 Report

Comments and Suggestions for Authors

The reviewer enjoyed reading the manuscript very much. Interesting idea to design an ear-level EEG sensor. Well conducted research and well composed papers. Kudos to the authors. Looking forward to seeing further progress. 

Author Response

We thank the reviewer for the positive comments and appreciate the recognition of the work. 

Reviewer 2 Report

Comments and Suggestions for Authors

The paper discusses the development of a toolbox for the assessment of Electroencephalography (EEG) enabled earbuds, aiming to enhance their characterization for broader health-related applications. The toolbox includes a desktop application called "EaR-P Lab" for integrating EEG validation paradigms and a novel approach using 3D ear scans (ear-EEG phantom) for controlled assessment of ear-EEG devices. The EEG paradigms were validated for scalp acquisitions and applied to ear-EEG measurements, demonstrating successful metrics for system characterization. The toolkit is designed to facilitate the systematic evaluation of novel ear-EEG devices, covering hardware to neural signals.

Since this is not an empirical paper, I can only comment on the quality of presenting this new technology to the EEG community. Except for some low quality images or images with very small text, which the authors should improve, I don't have any objections.

Author Response

We thank the reviewer for the appreciation of the manuscript and the work presented. 

We have vectorized the text in the figures with to improve the quality of the text within. 

Specifically :

- Figure 2 was replaced for a new and clearer version
- Figures 4, 5 and 29 were fixed/made slightly bigger
- The text size of Figures 6, 7, 22, 23, 27 and A4 to A7 was upscaled

Reviewer 3 Report

Comments and Suggestions for Authors

1. Do consider updating old references in the manuscript. Also, many websites are cited. Please ensure the cited websites are all necessary.

2. The manuscript is very comprehensive. I do enjoy reading the write-up. Nonetheless, I still think is good to have a Conclusion to summarize all the methodology and findings in one paragraph.

Author Response

We thank the reviewer for the comments and suggestions we have addressed the reviewer comments as follows:

  1. Do consider updating old references in the manuscript. Also, many websites are cited. Please ensure the cited websites are all necessary.
    1. We replaced reference 14 initially from 1998 with a newer reference supporting our arguments: Nordin, A.D.; Hairston, W.D.; Ferris, D.P. Dual-electrode motion artifact cancellation for mobile electroencephalography. Journal 828 of Neural Engineering 2018, 15
    2. We removed references to websites where in text citation was more appropriate. Website references 7, 26, 27, 30 and 35 were removed

2. The manuscript is very comprehensive. I do enjoy reading the write-up. Nonetheless, I still think is good to have a Conclusion to summarize all the methodology and findings in one paragraph.

  1. We have added a conclusion section summarizing the methodology and findings as suggested.